

# Cloud-to-Thing continuum-based sports monitoring system using machine learning and deep learning model

Amal Alshardan[1], Hany Mahgoub[2], Saad Alahmari[3], Mohammed Alonazi[4], Radwa Marzouk[5] and Abdullah Mohamed[6]

[1] Department of Computer Science, Princess Nourah bint Abdulrahman University, Riyadh, Saudi Arabia
[2] Department of Computer Science, King Khalid University, Riyadh, Saudi Arabia
[3] Computer Science, Northern Border University, Arar, Saudi Arabia
[4] Department of Information Systems, Prince Sattam bin Abdulaziz University, Al-Kharj, Saudi Arabia
[5] Department of Information Systems, Princess Nourah bint Abdulrahman University, Riyadh, Saudi Arabia
[6] Research Centre, Future University, New Cairo, Egypt

Corresponding author
Saad Alahmari,
saad.alahmari@nbu.edu.sa

## ABSTRACT

Sports monitoring and analysis have seen significant advancements by integrating cloud computing and continuum paradigms facilitated by machine learning and deep learning techniques. This study presents a novel approach for sports monitoring, specifically focusing on basketball, that seamlessly transitions from traditional cloud-based architectures to a continuum paradigm, enabling real-time analysis and insights into player performance and team dynamics. Leveraging machine learning and deep learning algorithms, our framework offers enhanced capabilities for player tracking, action recognition, and performance evaluation in various sports scenarios. The proposed Cloud-to-Thing continuum-based sports monitoring system utilizes advanced techniques such as Improved Mask R-CNN for pose estimation and a hybrid metaheuristic algorithm combined with a generative adversarial network (GAN) for classification. Our system significantly improves latency and accuracy, reducing latency to 5.1 ms and achieving an accuracy of 94.25%, which outperforms existing methods in the literature. These results highlight the system's ability to provide real-time, precise, and scalable sports monitoring, enabling immediate feedback for time-sensitive applications. This research has significantly improved real-time sports event analysis, contributing to improved player performance evaluation, enhanced team strategies, and informed tactical adjustments.

## INTRODUCTION

A rising number of people have shown an interest in the study of sports recordings that have been collected by a variety of cameras throughout the course of the last several years. A wide range of applications are made feasible as a result of the analysis. Some examples of these applications include the improvement of sports video broadcasts (*Akan & Varlı, 2023*; *Nuriddinov, 2023*), the reconstruction of 3D matches (*Lu et al., 2019*;

*Miao & Ge, 2023*), the provision of interactive content for audiences (*Amosa et al., 2023*), and the collection of game data to assist coaches in making tactical analysis (*Zaman et al., 2024*). When trying to get the maximum possible degree of knowledge of a sporting event, one of the most important activities to engage in is monitoring players. Regarding sports films, it is conceivable to conceive of player tracking as a multi-target, multi-camera tracking (MTMCT) assignment. This study aims to determine the position of every target at all times and to construct multi-camera trajectories from several different video streams (*Wang et al., 2018*). The players often swap positions in the eye of the camera, which results in the identity transition, which is one of the most critical aspects that greatly influences the accuracy of MTMCT. This transition is brought about by the players frequently exchanging locations. As a result, it is essential to control the identity changes to guarantee reliable monitoring results. In an attempt to find a solution to this problem, several different strategy methods include the development of features. It is necessary to define a color model for the drifting problem. Data about the game's context was compiled, and the motion likelihood of the participants was broken down into separate models for each individual participant for the whole game. The patch-based appearance model and the spatial-temporal similarity evaluation are used to aim for adaptive pedestrian tracking. On the other hand, these approaches are solely dependent on the custom-crafted appearance models, which cannot differentiate between players because some of their colleagues have a similar appearance. In light of this, to solve the problem of identity shifts, we will continue investigating the unique representation of the player identity *via* the utilization of deep learning. It is substantially more difficult to identify the participants in sports films based on real-world athletes than in conventional sports videos. Compared to pedestrians and autos, which have relatively predictable motion patterns, players tend to confound their opponents by making quick changes in direction and unexpected fluctuations in velocity. This contrasts with the fact that players tend to confuse their opponents. However, when it comes to player identification, characteristics often employed for re-identification, such as color and stride, are made ineffectual. This is the case. On the other hand, this is significantly different from the scenario in which human re-identification is carried out (*Zhang et al., 2019*)

In addition to the challenges of real-time analysis and low latency, ensuring data privacy and security is crucial, given the sensitive nature of player information being processed within the proposed system.

## Aim and scope

This research aims to develop a comprehensive sports monitoring system that seamlessly integrates cloud and edge processing for real-time analysis. The scope encompasses data acquisition from various sources, preprocessing for quality enhancement, feature extraction for essential information capture, cloud-based processing for in-depth analysis, continuum paradigm integration for localized processing, and decision-making based on integrated analysis. The system focuses on basketball sports events and aims to improve player performance evaluation, team strategies, and tactical adjustments.

### Research objectives

1) To acquire sports video data from diverse sources including live feeds, recorded matches, and archival footage.
2) To preprocess acquired data for noise reduction, frame alignment, and resolution adjustment.
3) To extract relevant features using ML and DL techniques for player, action, and event identification.
4) To perform cloud-based processing for in-depth analysis and application of machine learning algorithms for activity recognition and event detection.
5) To integrate edge devices within the continuum paradigm for localized processing, ensuring real-time analysis and reducing latency.
6) To make informed decisions regarding player performance, team strategies, and tactical adjustments using fuzzy decision-making techniques.

## LITERATURE REVIEW

*Tuncer et al. (2020)* presents a new local descriptor approach to feature extraction called multikernel local diamond pattern (MK-LDP). The goal of the proposed MK-LDP is to extract unique characteristics from a picture or signal by using multikernel functions and diamond-like patterns in both the vertical and horizontal directions. One may find ternary, quaternary, and signum kernels among them. One approach to extracting features for human activity identification (HAR) is MK-LDP. Proposed HAR technique consists of four main steps: preprocessing, feature generation using MK-LDP, informative features selection with hybrid ReliefF and neighbourhood component analysis (RFNCA), and classification with support vector machine (SVM). The suggested MK-LDP takes the raw sensor signals and extracts 2,560 features; RFINCA then uses these 2,560 features to choose the 512 most relevant ones, and finally, the SVM for HAR uses these 512 features as input.

*Tanberk et al. (2020)* suggested design is built by integrating deep learning techniques with supplementary movement information in video datasets and a dense optical flow methodology. This is the first research we know that uses a unique mix of optical flow-fed 3D-convolutional neural networks (3D-CNNs) and auxiliary information-fed long short-term memory networks (LSTM) across video frames to identify human activities. There are six main points made by this study. To start, the motion vectors are calculated using a 3D-CNN, also known as multiple frames. Dense optical flow, the distribution of apparent movement velocities in collected imagery data in video frames, is another application of 3D-CNN that serves the same aim. Thirdly, the LSTM is used as supplementary data in video for object and hand-tracking recognition. Fourth, video categorization is carried out using the support vector machine approach. Fifth, to show how the proposed research contributes, various comparison tests are run on two newly developed chess datasets, the magnetic wall chess board video dataset (MCDS) and the regular chess board video dataset (CDS). *Ronald, Poulose & Han (2021)* present iSPLInception, a deep learning model inspired by Google's Inception-ResNet architecture, which achieves high predictive

accuracy and reduces device resource consumption. We test the model on four public HAR datasets from the UCI machine learning repository and compare its performance to previously proposed DL architectures for the HAR problem. We find that the proposed model outperforms the existing approaches on multiple metrics of accuracy, cross-entropy loss, and F1 score across all four datasets. *Mekruksavanich & Jitpattanakul (2021)* suggested deep learning (DL), a subfield of machine learning that relies on complex artificial neural networks, has shown to be quite effective in recognition tasks according to S-HAR. Recent years have seen the effective application of many DL algorithms to the S-HAR task, including convolutional neural networks (CNNs) and recurrent neural networks (RNNs). Here, we zeroed in on four DL models—long short term memories (LSTMs), bilateral long short term memories (BiLSTMs), gated recurrent units (GRUs), and bidirectional gated recurrent units (BiGRUs)—based on RNNs that were able to handle challenging activity identification jobs. Additionally, four hybrid DL models were evaluated for efficiency; these models combined efficient RNN-based models with convolutional layers. Results from experiments conducted on the UTwente dataset showed that the proposed hybrid RNN-based models outperformed the state-of-the-art in recognition and several other performance metrics. *Lee et al. (2020)* of this research was to compare deep learning's performance in squat posture categorization to that of traditional machine learning. Furthermore, the ideal spot to install the sensors was identified. A total of 39 healthy individuals had five inertial measuring units (IMUs) fastened to their lumbar, left, right, and calf regions, respectively, to record their acceleration and gyroscope readings. Each person did six proper squats and five beginner-friendly variations of the exercise before moving on to the next. *Mekruksavanich & Jitpattanakul (2020)* provide a HAR framework that uses geo-temporal characteristics automatically retrieved from data collected by wristwatch sensors. The system uses a hybrid deep learning technique, using convolutional neural networks and long short-term memory networks, eliminating the need for human feature extraction. Additionally, Bayesian optimisation is beneficial in tuning the hyperparameters of all the networks under consideration. *Afsar et al. (2023)* suggests a game-based solution for physical fitness *via* wearable sensors, but it also suggests a versatile system that can be trained for other applications utilizing the domain-specific dataset. Many fields, including online education, sports, healthcare, fitness, and criminal detection, may benefit from the critical duties of gesture recognition and virtual reality portrayal. Specifically, the suggested system lets the user execute, identify, and portray various movements inside the VR game. The technique begins by removing outliers from the input data using a median filter. *Alghamdi (2023)* present a new football player health prediction method that uses wearable gear and recurrent neural networks. One of the earliest uses of wearable sensors for athletes' fitness and health, the suggested system tracks the players' vitals in real time. The time-step data is fed into a recurrent neural network, and from that data, deep features are extracted, allowing for the provision of health prediction results. In this experiment, data collected regarding the players' health dictates the results of several trials. *Hnoohom, Mekruksavanich & Jitpattanakul (2023)* To forecast physical activities, we provide a novel approach to feature extraction from the PPG signal using deep learning (DL). Our study aimed to find a way to effectively detect many sorts of

everyday activities from the raw PPG signal. To that end, we created a deep residual network called PPG-NeXt, built on convolutional operation, shortcut connections, and aggregated multi-branch transformation. Using just PPG data on the three benchmark datasets, the suggested model attained an experimental prediction F1-score of above 90%. *Biró et al. (2023)* To forecast physical activity essential performance indices in performance sports using machine learning (ML), this research examines the synergy possibility of data from medical radar sensors and tri-axial acceleration sensors. One unique aspect of this approach is using a 24 GHz Doppler radar sensor, which can detect vital signs like breathing and heart rate without physical contact, in conjunction with acceleration data from 3D accelerometers, to forecast the level of physical exertion. The data used in this study comes from professional athletes as well as publicly accessible datasets that were developed for academic research. The heart rate (HR) was measured using a medical radar sensor that uses no-contact remote sensing, and the activity's velocity was determined using 3D acceleration, both of which were part of the sensor data management system (*Xiao et al., 2023*). Due to the high need for both online and offline data, motion, and activity detection have made great strides in sports event recording. We use deformable learning techniques to improve traditional deep learning models for sports behavior detection and analysis. The method is an excellent choice for sophisticated frameworks that recognize sports recordings since it is efficient, resilient, and uses statistical analysis. A thorough grasp of action recognition is crucial for sports management and identification. This research presents a hybrid deep-learning architecture that can accurately classify human activities and sporting events.

Applications such as online gaming, real-time video conferencing, or autonomous vehicle control require extremely low delays in data transmission (latency) (*Sun et al., 2018*, *2015*). This process needs to be efficient, as recruiting too many users can be costly (*e.g.*, in terms of incentives or bandwidth usage), while too few or the wrong users can lead to inadequate data (*Wang et al., 2017*, *2023a*). In lower limb exoskeletons, impedance control is critical because it adjusts the level of support provided by the exoskeleton based on the user's movements (*Wang et al., 2024*; *Sun et al., 2024*). A neural network learns the implicit fields along the rays, predicting the depth and surface geometry more accurately (*Shi et al., 2023*; *Zhou et al., 2024*). These robots have potential applications in medicine, environmental monitoring, and industry, where their miniature size and upstream motility could be used for targeted interventions in complex (*Wang et al., 2023b*; *Ariyanto et al., 2023*). The concept of "body extension" refers to expanding the functional reach, skill, and versatility of a robotic system beyond what a single manipulator or fixed robotic arm can achieve (*Hirao et al., 2023*; *He et al., 2023b*). The RGAN first generates potential regions of the person's body by analyzing the input image (*He et al., 2023a*; *Luo et al., 2022*). The research examines how to translate the beetle's claw-like structure and adhesive system into a robotic design that can attach to surfaces securely in space environments (*Shi et al., 2024*; *Gu et al., 2024*). DialogueINAB captures how the emotional tone shifts between interlocutors by creating representations that reflect their attitudes and behaviors in each part of the conversation (*Ding et al., 2023*).

Implementing deep learning and wearable technologies for tracking athletes has gained considerable interest over the last decade. For instance, _Zhang (2021)_ evaluated several deep learning models for smart wearables, focusing on human movement recognition while using wearables in sports. This opens up possibilities for real-time assessment and injury prevention. _Ascioglu & Senol (2020)_ constructed a multi-sensor monitoring system that is wearable and wireless and capable of activity classification and recognition through the use of deep learning, and this was achieved with a very high level of accuracy in a variety of movement recognitions. _Cust et al. (2019)_ summarized the state of art of machine and deep learning approaches dedicated to recognizing movements related to specific sports and stressed the role of effective models and training data in such approaches. _Pajak et al. (2022)_ presented an innovative model supporting the recognition of sports activity based on inertial sensors and deep learning, which effectively identified complex activity. _Kautz et al. (2017)_ leveraged advancements in deep convolutional neural networks to recognize activities while playing beach volleyball, demonstrating the viability of applying deep learning to sports with fast and unsupervised settings. _Wang, Cui & Fan (2023)_ built a wearable-based sports health monitoring system employing a hybrid model of CNN and LSTM with self-attentions, which was exceptionally useful in the long-term monitoring of the health and performance of the athletes. _Afsar et al. (2023)_ concentrated on assessing physical exercises in Exergaming with the help of deep learning and wearable sensors. Such systems are said to be helpful in activities across various sports. _Mekruksavanich & Jitpattanakul (2022)_ probed into the multimodal wearable sensing for sport-associated activity recognition and enhancement by applying deep learning networks for activity classification. Finally, the work of _Chang, Sun & Ali (2024)_ included utilizing a cloud-connected smart monitoring system based on SVM and CNN, reflecting the potential of cloud computing in the development of efficient and scalable systems for sports monitoring. All in all, the results of these researches further realize the deep learning and the use of wearable technologies in sports monitoring and evaluating performance.

### Overall problem statement

Sports monitoring systems often face real-time analysis, resource utilization, and decision-making challenges. Latency issues, data quality, and inadequate feature extraction techniques hinder accurate performance evaluation and strategic decision-making. Additionally, integrating cloud and edge processing seamlessly poses technical challenges. This research addresses these challenges by proposing an innovative system that leverages ML, DL, and fuzzy decision-making techniques to achieve real-time analysis, optimize resource utilization, and improve decision-making in sports monitoring.

## PRELIMINARIES

To elucidate the planned effort, this section uses background information on the Harmony Search Algorithm (HSA), a hybrid meta-heuristic algorithm, and the particle swarm optimisation method (PSO).

# HARMONY SEARCH ALGORITHM

The goal of the metaheuristics method known as the Harmony Search method is to enhance musical compositions by modifying the characteristics of individual instruments to get a more harmonious overall sound. Based on mathematical analysis, the HSA outperforms previous optimisation methods and handles a broad range of optimisation issues. Among the many types of technical optimisation issues that the HSA method may solve are those involving distributed networks, routing, and puzzle solving. Three phases typically comprise an HSA: startup, improvisation, and updating. These processes may be expressed as,

$$H_i^{New} = \begin{cases} H_i(l) \in \{H_i(1), H_i(2), \ldots, H_i(l)\} R_1 > hmcr \\ H_i(l) \in \{H_i^1, H_i^2, \ldots H_i^{HMS}\} R_1 \le hmcr \\ H_i(l) + R_3 * BW \quad R_2 \le par. \end{cases} \tag{1}$$

The new harmony $H_i^{New}$ is represented by the variables $R_1, R_2, and\ R_3$, which are random values between 0 and 1. The variables hmcr, par, and BW stand for the pitch adjustment rate and bandwidth, respectively.

The accuracy of HSA algorithms is often compromised because to their fast convergence rate. So, to improve the convergence rate, various changes were made to the HSA algorithm, such as dynamically adapting the HSA parameters. Enhanced workflow precision by integration with additional optimisation methods (*i.e.*, hybrid solutions). New harmony's established norms (sometimes called "Hybrid harmony").

## Particle swarm algorithm

The stochastic optimisation technique known as the particle swarm algorithm takes its cues from the population dynamics of many animal groups, including schools of fish and flocks of birds. There are two primary study methods that are often associated with PSO algorithms, and these are evolutionary algorithms and artificial algorithms. In terms of optimisation, this technique outperforms genetic optimisation approaches. This operation is carried out using the velocity vector and the swarm size of each particle's location vector in a finite dimension. From the starting point, the swarm experiences the ideal positions of each person and updates their optimal positions. Revising, taking into account both position and velocity, the ideal location of each individual using swarm. As an expression of the position update formula,

$$y_i^{q+1} = y_i^q + P_i^{q+1}. \tag{2}$$

Iteration is represented by q, particle by i, the particle's location by $y_i^q$, and the particle's velocity by $P_i^q$. And to top it all off, the following is the updated velocity formula that takes inertia weight into account:

$$P_i^{q+1} = s * [P_i^q + l3_1\eta_1(R_i^q - y_i^q) + l3_2\eta_2(R_i^q - y_i^q)]. \tag{3}$$

The current stored velocity is represented by $l3_1\eta_1$ and $l3_2\eta_2$, the iteration's pbest and gbest are $R_i^q$ and $R_i^q$, the social factors of the stochastic algorithm are $\eta_1$ and $\eta_2$, and the

weight factor of inertia is s. It gets around the problems with the HSA algorithm. Compared to other heuristic optimization methods, this one offers superior computing efficiency because of its efficient control settings.

It circumvents the problems with the HSA algorithm. Because of its efficient control settings, this method offers superior computing efficiency compared to other heuristic optimization methods.

## PROPOSED METHODOLOGY

The block diagram, as shown in Fig. 1, visually represents the methodology of the proposed Cloud-to-Thing continuum-based sports monitoring system. It begins with data acquisition from multiple sources, such as cameras and sensors, followed by preprocessing to enhance data quality. The next stage involves feature extraction using advanced methods like improved Mask R-CNN. The extracted features are then processed in the cloud using deep learning models. The continuum paradigm integration combines cloud and edge processing results to optimise real-time performance. Finally, the decision-making stage utilises integrated analysis for player performance evaluation and tactical adjustments. This diagram provides a clear overview of the system's data flow and processing stages.

### Data acquisition

The first step is to gather video data related to sports from several sources, like live streams, recorded contests, or archive material. Additional information, such as game outcomes, ambient factors, and player stats, may also be included in the data.

Dataset: The APIDIS dataset contains seven camera images and is open to the public. The video files are captured in MPEG-4 at a frame rate of 25 fps with a resolution of 800 × 600, and they include 1,500 frames. There are 16 unlabeled periods in the dataset as well. The dimensions of the court for basketball are 2,797 cm × 1,499 cm. Two officials, two five-player teams, and a total of twelve players make up the court. Lighting, reflections, and shadows make the publicly accessible dataset difficult to work with. Our group at Shantou University has just amassed a new dataset, which we call the STU dataset. There are eight cameras in this time-synchronized dataset. The MPEG-4 video files are captured at a frame rate of 24 fps with a resolution of 1,280 × 720. Two time periods, totaling 2,500 frames, have been used in our investigations. The basketball court's dimensions are 2,800 cm × 1,500 cm. Two referees and two teams of eight make up the sixteen players on the court. Eleven whole periods make up the dataset. We plan to release it at a later date.

The basketball court dimensions used in this study must align with standard measurements for accurate analysis and application. In the National Basketball Association (NBA), the standard court size is 94 feet by 50 feet (approximately 28.7 m by 15.2 m). Under the rules of the International Basketball Federation (FIBA), the court is slightly smaller, measuring 28 m by 15 m. To ensure consistency with these standards, the system in this study has been calibrated to support data collection and analysis based on these official court dimensions. This standardization facilitates the applicability of the proposed system in professional and international basketball settings.

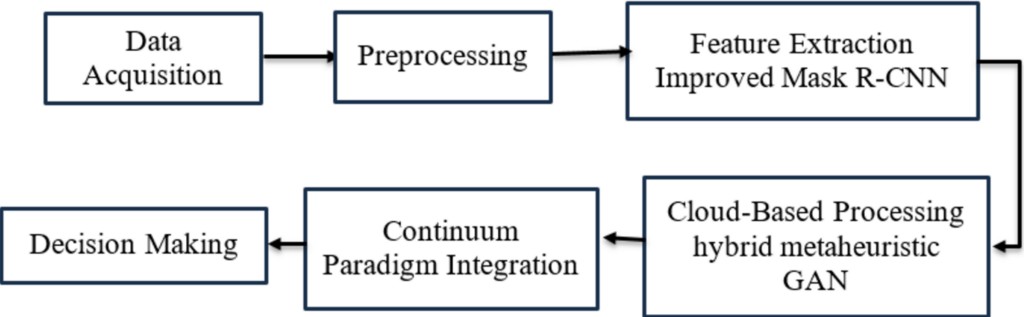

**Figure 1 Overall processing flow of the proposed methodology.**

This research follows ethical norms with great respect regarding data collection and usage, which is very important because player data is delicate. Before any data collection, an ethical review, for instance, by an Institutional Review Board (IRB) or any appropriate ethical review body, is performed to evaluate and accept the study design. Specifically, all subjects who participated, including athletes and athletic trainers, were oriented with the reason, extent, and localization associated with the utilization of gathered information. Clear instructions on how to pursue ethical practices concerning voluntary participation are given to everyone involved in the investigation. Special identification restrictions are imposed wherever applicable, especially in the preprocessing period, to protect participant information from being traced to specific persons. Further, only authorized personnel have access to the data, and safe data storage methods are followed to avoid misuse of the information. The research also considers local legislation on protecting data collected during the study, which applies to all the studies that intersect the boundaries of the European Union or other regional laws. These measures protect and journalist all participants in the research and the rights of all participants involved in the research which helps the advancement of ethics in the research and development of the sports monitoring system.

## Preprocessing

The collected data is preprocessed to improve the quality and make further analysis easier. Adjusting the resolution, stabilizing the video, reducing noise, and aligning the frames are all examples of preprocessing operations. The first step is to use the threshold to filter out the noise. The image's noise is also eliminated to improve the picture quality. To achieve this goal, we put forth the Thresholding Adaptive median filter (TAMF), which can remove noise and produce point clouds of excellent quality.

The noisy picture is denoted as $\check{K}(l)$, while the noisy word is $\check{K}(r)$. Here is the representation of these inputs in formulation form:

$$\check{K}(l)= S(o) + N(8) \tag{4}$$
$$\check{K}(r)= S(o) + N(8). \tag{5}$$

In this case, $S(o)$ stands for the initial data and $N(8)$ for the background noise. The noise reduction process begins with converting the photos to greyscale. The median

filtering is subjected to a sliding window with dimensions ($\partial = v \times v$). The RGB-D picture's pixels, which are ranked from tiny to big according to their grey value, are represented by the symbols $\check{K}(\eta)$. The number that represents the median of a pixel is $\check{K}_{med}$.

With $\check{K}_{min}$ being the lowest grey value in the window and $\check{K}_{max}$ being the highest, we can express the grey value of the centre pixel as $\check{K}_\zeta$. The original information of the picture $\check{K}_\zeta$ is characterised as $\zeta$ if it is not equal to $\check{K}_{min}$ or $\check{K}_{max}$; otherwise, it is regarded as noise. To accomplish denoising, replace $\check{K}_\zeta$, with $\check{K}_{med}$.

The adaptive adjustment is made to the window size so that the filtering template may be larger. The symbol $\check{K}(\xi)$ represents the pixel value in this filter window. The two stages of adaptive median filtering are as follows:

**Step 1:** If $\check{K}_{min} < \check{K}_{med} < \check{K}_{max}$ shift to step 2 otherwise, adjust the window as $v = v + 2$. When $\partial > \partial_{max}$, then the output is $\check{K}_{med}$ otherwise, repeat step 1.

Where, $\partial_{max}$ denotes the window's upper limit.

**Step 2:** If $\check{K}_{min} < \check{K}_{med} < \check{K}_{max}$ then the output is $\check{K}(\xi)$ otherwise $\check{K}_{med}$.

Step 2 doesn't save borders and other information that are crucial for picture smoothing, and it may lower the window's upper limit. To reduce the residual noise while maintaining the edge and other crucial information, the enhanced wavelet threshold function is used. There are two kinds of frequency coefficients in this threshold function: low-frequency coefficients, which include image information, and high-frequency coefficients, which contain noise and edge information. Applying the wavelet cutoff function to high-frequency coefficients helps to remove noise while preserving relevant information. There are two types of threshold functions that are taken into account: soft and hard. A representation of the hard threshold function would be,

$$h_{th} = \begin{cases} h & |h| \geq t \\ 0 & |h| < t \end{cases} \tag{6}$$

where $h_t$ denotes the wavelet threshold. The soft threshold function is represented as,

$$h_{ts} = \begin{cases} sign(h)(|h| - t), |h| \geq t \\ 0, |h| < t. \end{cases} \tag{7}$$

At $|h| = t$, the enhanced wavelet threshold function becomes continuous, and it gains the overall definition domain with different sorts of slopes from $|h| < t$ and $|h| \geq t$. This is the revised wavelet threshold function put into formula form:

$$h_{ti} = \begin{cases} sign(h)(|h| - t), |h| \geq t \\ (1 - \lambda) |h|, |h| < t \end{cases} \tag{8}$$

where $\lambda$ stands for the factor that may be adjusted. During the interval $|h| < t$, the threshold function is not equal to zero because of the adjustment factor $\lambda(0 < \lambda < 1)$. Although the soft threshold function removes the majority of the noise from pictures and noisy points in point clouds, this demonstrates that the wavelet coefficients are somewhat less compressed. To increase detection accuracy, we convert points to voxels at the second level using cloud points to provide a better perceptual picture. The three coordinates

**Algorithm 1 Noise removal algorithm.**

1: Initialize $\{l, r, \xi\}$

2: Initialize Kmin, Kmed, Kmax

3: **for** $l$, $r$ **do**

4:     **if** the step 1 condition is not met **then**

5:         **if** $\partial > \partial_{\max}$ **then**

6:             Repeat step 1

7:         **else**

8:             Shift to step 2

9:         **end if**

10:     **end if**

11:     Compute $h_{\text{th}}, h_{\text{ts}}$ using Eqs. (3) and (4)

12:     Compute $h_{\text{ti}}$ and adjust threshold using Eq. (5)

13: **end for**

14: Return denoised $l$, $r$

$(X, Y, Z)$ that each cloud points to have three sets of corresponding values, such as $([x_{min}, \, x_{max}], \, [y_{min}, \, y_{max}], \, [z_{min}, \, z_{max}])$. The point cloud data is first organised into voxel subsets according to the Cartesian coordinate system, with each subset being represented by an index $U(d, f, c)$.

Where $d \in [0; L_x - 1]$, $f \in [0; L_y - 1]$, and c $c \in [0; L_z - 1]$. The count of voxels $(\Delta_x, \, \Delta_y, \, \Delta_z)$ in each direction is given according to the individual voxels' dimensions $(L_x, \, L_y, \, L_z)$.

$$L_x = \frac{(x_{max} \, - \, x_{min})}{\Delta_x} + 1 \tag{9}$$

$$L_y = \frac{(y_{max} \, - \, y_{min})}{\Delta_y} + 1 \tag{10}$$

$$L_z = \frac{(z_{max} \, - \, z_{min})}{\Delta_z} + 1 \tag{11}$$

In this context, $\Delta_x, \, \Delta_y, \, \Delta_z$ indicate the voxel sizes, while $L_x, \, L_y, \, L_z$ stand for the voxel counts from left to right. The following description of pseudocode serves as a concise explanation of noise reduction (Algorithm 1).

## Feature extraction

The next step is to extract important characteristics from the cleaned data in order to record crucial details about the participants, their activities, and the events that took place. Image processing, object identification, motion estimation, and posture estimation are some of the approaches that feature extraction may use. We generate region boxes using improved Mask R-CNN (IMR-CNN), which helps to achieve high accuracy of pose

estimate by eliminating the misalign issue of classic Mask RCNN, and we consider both point-based and local-based semantic information for semantic segmentation.

In classic Mask R-CNN, the classification head is responsible for ROI classification and bounding box regression. A quicker R-CNN-based instance segmentation approach, the improved Mask-R-CNN is an expansion of that model. A modification to the mask's head block allows for more accurate boundary identification, which in turn improves the mask-RCNN. The first step in learnable upsampling is to add a decoder layer to feature maps; this effectively increases the spatial resolution. The second step is for the ROI align block with skip connections to align the ROI. This will provide features with high resolution, which the adjusted mask head may then use.

### Improved mask R-CNN network structure

Segmenting the combined picture using the enhanced Mask-R-CNN technique improves the precision of 3D object recognition. It has an input block, a backbone network, a feature pyramid network (FPN), two kinds of heads (class and mask heads), and a ROIAlign function for alignment. Locations and objects are identified by the RPN, bounding box classification is done by the class head, and masks are extracted pixel-wise from the cropped features obtained from the bounding box using the mask head.

### RPN network

To determine the bounding boxes around individual objects and to identify each object class, the RPN block takes the picture characteristics as input. While low-resolution feature maps are used to extract big items, high-resolution feature maps are utilised to extract tiny things.

### Backbone network

A ResNet101 hybrid deep encoder model with FPN is used in the backbone network. Mast R-CNN uses this hybrid backbone network to extract feature maps from input photos of different sizes and resolutions. By combining FPN with ResNet *via* lateral connections, top-down routes, and bottom-up pathways, multi-scale semantic characteristics (*i.e.*, local and point) may be retrieved. For the sake of argument, let's pretend that the residual building block for residual mapping is

$$B = F(Y, \{\Psi_i\}) + Y. \tag{12}$$

The left-hand residual mapping that will be learnt is shown as $F(Y, \{\Psi_i\})$, where B is the input vector and Y is the output vector. In order for the dimensions of F and Y to match using shortcut connections, which are represented as, a linear projection $\Psi_s$ must be done, unless they are identical in the given equation.

$$B = F(Y, \{\Psi_i\}) + \Psi_s Y. \tag{13}$$

Then the feature pyramid is viewed based on image pyramid. In general, ROI assigning with h $x$ m to $r_q$ of the FPN which is represented as,

$$q = \left\lceil q_0 + log_2 \left( \frac{\sqrt{wh}}{360 \times 640} \right) \right\rceil \qquad (14)$$

where r denotes the feature pyramid, $q_0$ represents the target level of ROI with $h \times w$ which are mapped and $360 \times 640$ represents the height and width of the image size.

### ROI alignment

The ROI bounding box in Fast R-CNN uses the ROI pooling layer block to get the feature map from the small-sized backbone network. The Fast R-CNN uses the max-pooling layer to reduce the amount of the ROI features into a tiny, stable spatial extent of $h \times m$ feature map. In this case, $h \times m$ stands for the layer's hyperparameters, which are their height and width. Feature extraction using the spatial quantization approach causes the segmentation misalignment. To get around these problems while enlarging the ROI, you may use the ROIAlign block with dimensions of $(14 \times 14)$ pixels, which cancels the rounding procedure. The second ROIAlign layer, which is $56 \times 56$ pixels in size, is responsible for generating high-resolution features for the adjusted mask head. The RPN employs bilinear interpolation $\omega$ and generates anchor points using $\Psi$ integers. For the sake of argument, let's say that the four-digit anchor points used to locate object pixels are written as

$$\Psi = \{A_1, A_2, A_3, A_4\} \qquad (15)$$

where, the base points are represented as $\omega$ which are expressed as follows,

$$
\begin{aligned}
ROI(\mathrm{H}, 3) \approx & \frac{ROI(A_1)}{(H_2 - H_1)(H_2 - H_1)} * (H_2 - 3)(H_2 - 3) \\
& + \frac{ROI(A_2)}{(H_2 - H_1)(H_2 - H_1)} * (H_2 - 3)(H_2 - 3) \\
& + \frac{ROI(A_3)}{(H_2 - H_1)(H_2 - H_1)} * (H_2 - 3)(H_2 - 3) \\
& + \frac{ROI(A_4)}{(H_2 - H_1)(H_2 - H_1)} * (H_2 - 3)(H_2 - 3)
\end{aligned}
\qquad (16)
$$

where, H and 3 denote the directions (*i.e.*, axis).

### Loss function

A loss occurs in Mask-RCNN as a result of learning for many tasks in ROI sampling. Mask loss accumulation, classification loss, and bounding box loss (shown as), among others, are the outcomes of this.

$$\pounds = \pounds_m + \pounds_{\acute{c}} + \pounds_b. \qquad (17)$$

The loss of mask prediction during segmentation is represented by, $\pounds_m$, the loss of class-label prediction by $\pounds_{\acute{c}}$, and the loss of bounding-box refinement by $\pounds_b$. The given equation demonstrates that object segmentation is executed with high generalizability.

Pose estimation is carried out after object segmentation. Because we need to estimate the posture of objects in 3D object recognition, which is difficult because occlusion makes the procedure more difficult because the pictures' poses are different. In order to

estimate the pose from the bearing angle, the affine transformation is used, which involves translating and rotating the boxes relative to their location and orientation. Finding three-dimensional objects that are highly related to one another in successive frames from the past and the present. The process of estimating an object's location by computing its bearing angle by locating its centroid. Once all affine samples have been collected, use the current 3D picture frame to render the object area. Next, make sure the object's size is normal, and use bearing angles to guess the 3D objects' poses.

## Cloud-based processing

For further processing and analysis, the extracted characteristics are sent to the cloud. The use of deep learning and machine learning algorithms in cloud computing allows for the analysis of player motions, identification of activities, and performance evaluation. Activity identification and event detection may be achieved using techniques like GAN with a hybrid metaheuristic algorithm. The fundamental concept of GANs is the rivalry between two neural networks. Proposed architecture is shown in Fig. 2. The generator and discriminator are the two main components of GANs. The generator's job is to trick the discriminator into thinking the inputs are real, while the discriminator's job is to tell the difference between the two. Think of a neural network (generator) $gene(\mathfrak{Z}; \theta_{\mathfrak{G}})$ that maps the output sample of $(\mathfrak{Z} \sim \mathscr{P}_{\mathfrak{Z}})$ to the input samples of $(x \sim \mathscr{P}_{\mathfrak{G}})$. The output of this neural network called $Discr(x; \theta_{\mathfrak{D}})$, is a binary value that is determined by the inputs. The GAN's goal is to train both neural networks in tandem using each other's outputs; this training process is like a min-max game. The goal statement is,

$$\min_{gene} \max_{Dicr} \mathscr{V}(Discr, gene) = \mathbb{E}_{x \sim \mathscr{P}_{data}}(x)[\log Discr(x)] + \mathbb{E}_{\mathfrak{Z} \sim \mathscr{P}_{\mathfrak{Z}}}(\mathfrak{Z})[\log(1 - Discr(gene(\mathfrak{Z})))] \quad (18)$$

The loss function of GANs is among the distributions of $\mathscr{P}_{\mathfrak{Z}}$ and $\mathscr{P}_{\mathfrak{G}}$ respectively based on the kullback-Leibler distribution. The loss function can be formulated as,

$$Discr_{kl}(\mathscr{P}_{\mathfrak{Z}} \| \mathscr{P}_{\mathfrak{G}}) = \int_{-x}^{x} \mathscr{P}_{\mathfrak{G}}(x) \log \frac{\mathscr{P}_{\mathfrak{G}}(x)}{\mathscr{P}_{\mathfrak{Z}}(x)} \, dx. \quad (19)$$

An algorithm that combines the HSA with the PSO is a hybrid metaheuristic. We need to update the HS parameters frequently using the PSO algorithm since the HSA method quickly slips into local optima. There are three stages to the proposed HSA: initiation, improvisation, and updating. In this case, improvision is revised for three procedures: random selection, pitch adjustment, and memory consideration. Harmony memory (HM) is a part of the HSA algorithm that stores the method's goal function in harmony vectors. First, set up the HM and swarm size to start with so you can evaluate the fitness function, which is stated as follows:

$$\begin{bmatrix} h_1^1 & h_2^1 & \cdots & h_d^1 \\ h_1^2 & h_2^2 & \cdots & h_d^2 \\ \vdots & \vdots & \vdots & \vdots \\ h_1^x & h_2^x & \cdots & h_d^n \end{bmatrix} \begin{bmatrix} f_1 \\ f_2 \\ \vdots \\ f_n \end{bmatrix} = \begin{bmatrix} F_1 \\ F_2 \\ \vdots \\ F_n \end{bmatrix} \quad (20)$$

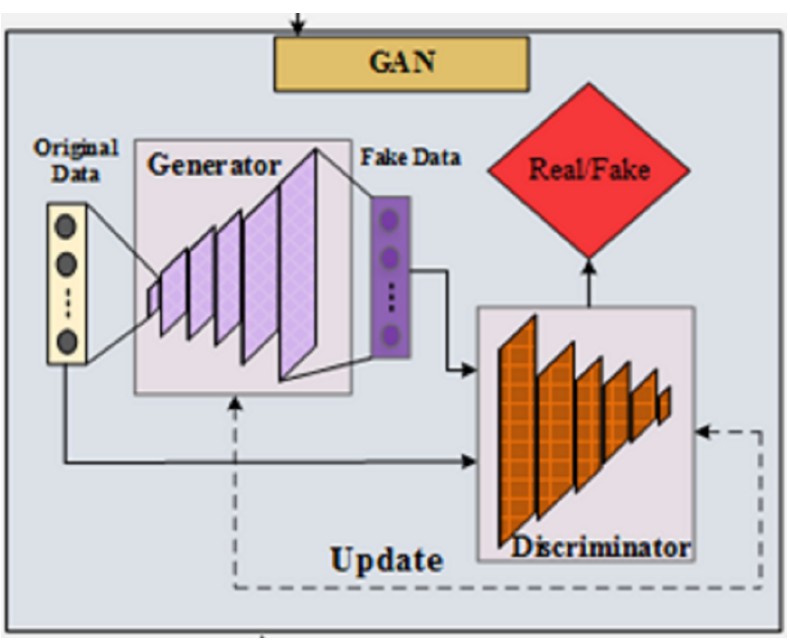

**Figure 2 Architecture of GAN.**

In this case, distance and location assess the fitness function. A fresh harmony vector $[h'_1, h'_1 \ldots, h'_t]$. is generated by performing harmonic improvisation once initialization is finished. A new harmony vector component $h'_j$ is created using the following formula.

$$h'_j \leftarrow \begin{cases} h'_j \in P_{HM} \text{ with Pb of } HMCR \\ h'_j \in h_j \text{ with Pb of } (1 - HMCR) \end{cases} \tag{21}$$

where HMCR stands for the component selection probability defined by the harmonic memory consideration rate, The formula for determining the pitch adjustment for the chosen $h'_j$ is given below.

$$h'_j \leftarrow \begin{cases} hn_j \in P_{HM} \text{ with Pb of } PAR \\ h'_j \text{ with Pb } (1 - PAR). \end{cases} \tag{22}$$

The parameter PAR stands for the pitch adjustment rate, $P_{HM}$ for the suggested hybrid HSA and PSO method, and $h'_j$ for the ideal controller position.

The goal function value for each $P\_Best$ is used to compute the new hybrid harmony vector. For $P_{HM}$ to take into account a new harmony vector, its objective value must be higher than that of the worst harmony. According to $P_{HM}$, the harmony vector with the poorest value is discarded. The best particles in the swarm, denoted as $P\_Best$, are used to identify the ideal location for the controller to be placed. It iteratively creates n particles, with the best one being named $G\_Best$. Every velocity updates the particle's location and velocity. Here is how the current location and velocity are defined:

$$v_{ij}(R+1) = W(R)v_{ij}(R) + C_1 R_1 \big(P\_Best_{ij} - P_{ij}(R)\big) + C_2 R_2 (G_{Best} - P_{ij}(R)) \tag{23}$$

$$P_{ij}(R+1) = P_{ij}(R) + v_{ij}(R+1) \tag{24}$$

**Table 1 HSA parameters.**

| Parameter names | Parameter symbols | Value |
|---|---|---|
| Harmony memory consideration rate | HMCR | 0.8 |
| Pitch adjusting rate | PAR | 0.2 |
| Random | Rand | 0.1 |
| HS iteration | H(I) | 100 |
| Harmony memory | HM | 5 |
| Minimum bandwidth | BW | 0.1 |
| Maximum bandwidth | MBW | 0.4 |

where $v_{ij}(R)$ and $P_{ij}(R)$ are the velocities and positions of the particles, $W(R)$ is the weight value, $C_1$ $and$ $C_2$ are the acceleration constants, $R1$, $R2$ are the random values between $[0, 1]$, and $P\_Best$ and $G\_Best$ are the local and global positions of the particles, respectively. The following is a definition of the weight value calculation:

$$W(t) = W_u - (W_u - W_l)\left(\frac{i}{I_{max}}\right) \qquad (25)$$

where represent the total count of iterations and $i$ represent the current iteration, and represent the upper and lower limit of the weight values. These processes are continued till met the termination criteria. Table 1 describes the parameters of HSA algorithm.

In this context, $I_{max}$ stands for the maximum number of iterations, i for the current iteration, and $W_u$ and $W_l$ for the maximum and minimum values of the weights, respectively. Until the termination requirements are satisfied, these procedures will be maintained. Figure 3 shows the process flow for hybrid optimization. Figure 4 illustrates the latency.

## Continuum paradigm integration

In parallel, edge devices within the continuum paradigm receive a subset of the data for localized processing.

Edge devices perform real-time analysis to provide immediate feedback and support time-sensitive applications. This integration ensures a balance between centralized cloud processing and distributed edge processing, optimizing resource utilization and reducing latency. The results from cloud-based and edge-based processing are fused and integrated to comprehensively analyze sports events. Fusion techniques combine insights from both sources to enhance accuracy, completeness, and reliability.

## Decision making

Based on the integrated analysis, decisions are made regarding player performance, team strategies, and tactical adjustments. Decision-making algorithms may consider player statistics, historical data, game context, and coach preferences. Evaluate the fuzzy weights

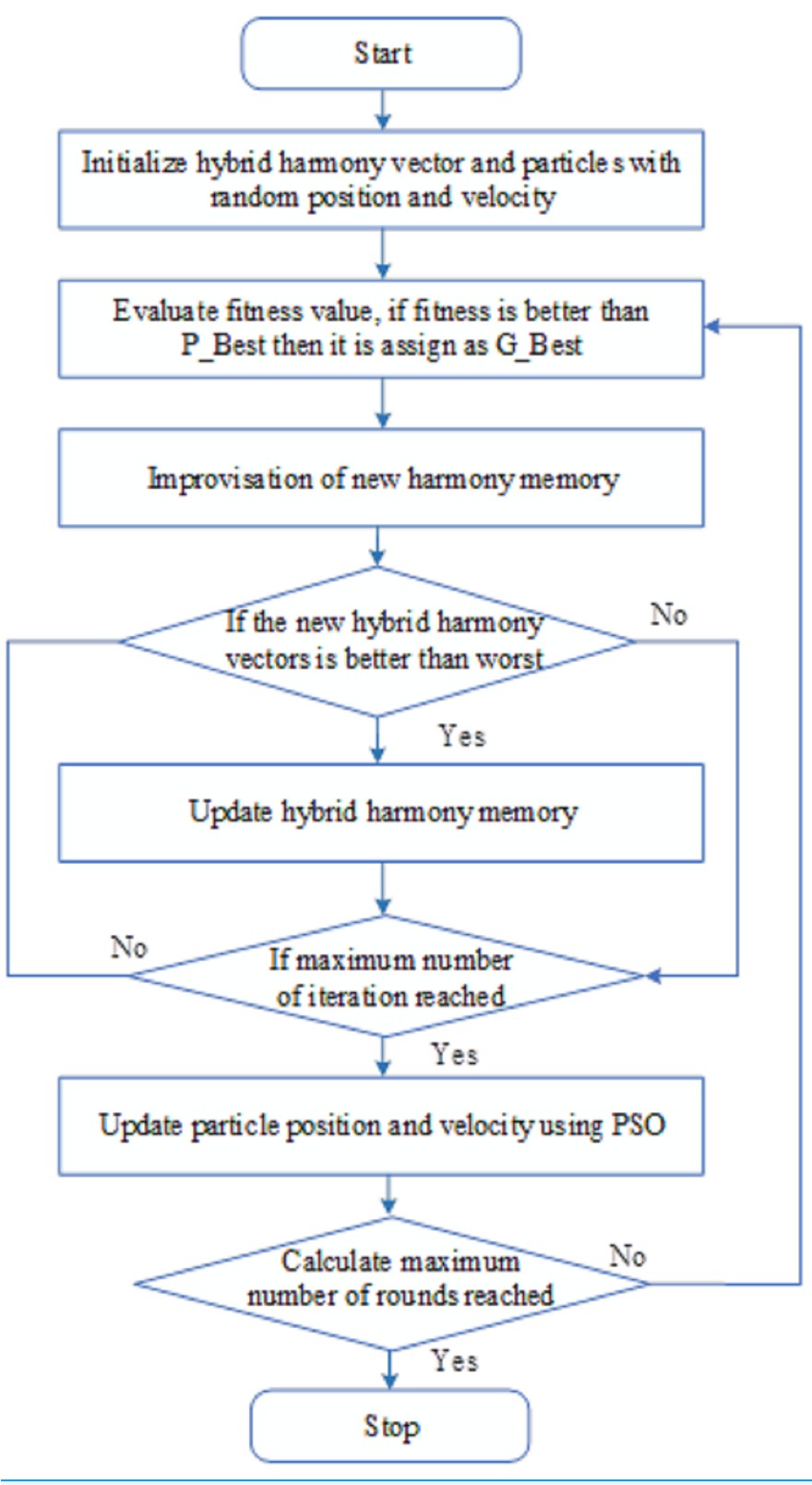

**Figure 3  Hybrid optimization.**               

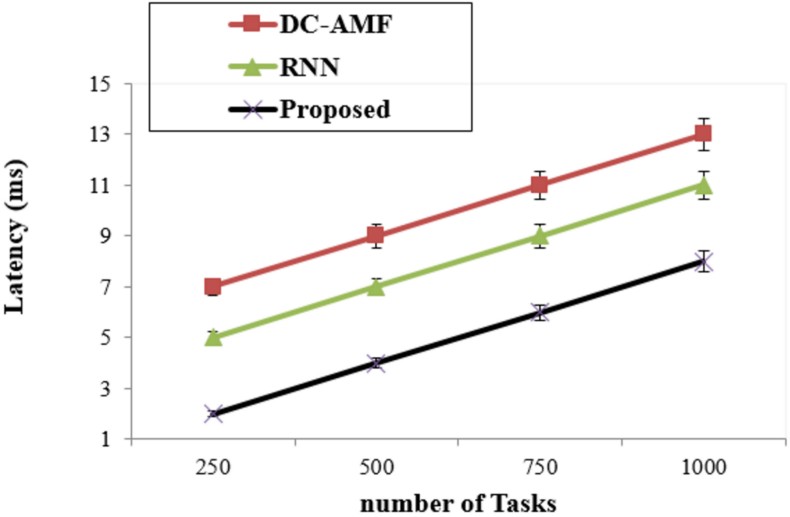

**Figure 4** **Latency.**

and fuzzy alternative rates by edge decision makers using sets of linguistics variables $(LV_2)$, which can be used to improve the fuzzy model by instances.

The fuzzy weight instances are Very low, low, medium-low, low, very high, medium-high, and medium, whereas the fuzzy alternatives instances are very poor, poor, fair, good, medium good, very good, medium poor, and good.

The decision matrix of fuzzy weights of the users and applications is represented as $w_h(h = 1, 2, \ldots, n)$, for the $r - th$ norm provided by the $h - th$ decision maker can be formulated as,

$$W^{rh} = [w_{1h} \ldots w_{rh} \ldots w_{lh}]. \tag{26}$$

The policy ratings of User (U) and Application (A) decision matrix can be represented as $UA_h(h = 1, 2, \ldots, n)$ for the $i - th$ alternative with respect to $r - th$ criterion provided by the $h - th$ decision maker can be represented as,

$$UA_h = \begin{bmatrix} ua_{11h} & \cdots & ua_{1rh} & \cdots & ua_{1lh} \\ \vdots & \vdots & \vdots & \vdots & \vdots \\ ua_{i1h} & \cdots & ua_{irh} & \cdots & ua_{ilh} \\ \vdots & \vdots & \vdots & \vdots & \vdots \\ ua_{b1h} & \cdots & ua_{brh} & \cdots & ua_{blh} \end{bmatrix}. \tag{27}$$

**Step 2: Construction and weighted matrix defuzzification**

The weighted policy rating matrix can be computed by multiplying each set of policy ratings $\{\exists^{irh} = (\exists^{irh1}, \exists^{irh2}, \exists^{irh3}, \exists^{irh4})\}$ with fuzzy weights $W^{rh} = W^{rh1}, W^{rh2}, W^{rh3}, W^{rh4}$ which can be calculated by,

$$\exists^W_{irh} = \exists^{irh} \times W^{rh} = (\exists^W_{irh1}, \exists^W_{irh2}, \exists^W_{irh3}, \exists^W_{irh4}). \tag{28}$$

The matrix form of weighted policy rating is represented by,

$$UA_h{}^D = \begin{bmatrix} ua_{11}^z & \cdots & ua_{1r}^z & \cdots & ua_{1l}^z \\ \vdots & \vdots & \vdots & \vdots & \vdots \\ ua_{i1}^z & \cdots & ua_{ir}^z & \cdots & ua_{il}^z \\ \vdots & \vdots & \vdots & \vdots & \vdots \\ ua_{b1}^z & \cdots & ua_{br}^z & \cdots & ua_{bl}^z \end{bmatrix}. \tag{29}$$

The normalization of $UA_h{}^D$ can be computed as follows,

$$\exists_{irh4}^{z+} = \max_i \{\exists^{irh4}\}, \; \textit{for benefit norm}$$

$$\exists_{irh1}^{z+} = \min_i \{\exists^{irh1}\}, \; \textit{for cost norm}.$$

The max, min function is used for differentiate the gathered fuzzy values. The normalized matrix can be represented as,

$$\overline{UA}_h{}^D = \begin{bmatrix} \overline{ua}_{11}^z & \cdots & \overline{ua}_{1r}^z & \cdots & \overline{ua}_{1l}^z \\ \vdots & \vdots & \vdots & \vdots & \vdots \\ \overline{ua}_{i1}^z & \cdots & \overline{ua}_{ir}^z & \cdots & \overline{ua}_{il}^z \\ \vdots & \vdots & \vdots & \vdots & \vdots \\ \overline{ua}_{b1}^z & \cdots & \overline{ua}_{br}^z & \cdots & \overline{ua}_{bl}^z \end{bmatrix}. \tag{30}$$

The defuzzification of normalized weighted policy rating matrix is normalized which can be represented as,

$$\overline{DF}\left(\overline{UA}_{ih}{}^D\right) = \frac{DF\left(\overline{UA}_{ih}{}^D\right)}{\sum_{L=1}^{b} DF\left(\overline{UA}_{Lh}{}^D\right)}, \; h = 1, 2, \ldots, n. \tag{31}$$

The normalized $\overline{DF}\left(\overline{UA}_{ih}{}^D\right)$ can be expressed as,

$$\overline{UA}_h{}^D = \begin{bmatrix} \overline{DF(ua_{11}^z)} & \cdots & \overline{DF(\overline{ua}_{1r}^z)} & \cdots & \overline{DF(ua_{1l}^z)} \\ \vdots & \vdots & \vdots & \vdots & \vdots \\ \overline{DF(ua_{i1}^z)} & \cdots & \overline{DF(\overline{ua}_{ir}^z)} & \cdots & \overline{DF(ua_{il}^z)} \\ \vdots & \vdots & \vdots & \vdots & \vdots \\ \overline{DF(ua_{b1}^z)} & \cdots & \overline{DF(\overline{ua}_{br}^z)} & \cdots & \overline{DF(ua_{bl}^z)} \end{bmatrix}. \tag{32}$$

Algorithm 2 computes fuzzy weights, normalizes decision matrices, and applies defuzzification to aid in decision-making processes.

## Data privacy and security considerations

Protecting susceptible player information is paramount in the designed Cloud-to-Thing Continuum-based Sports Monitoring System. To ensure that the data's confidentiality and integrity are not compromised, various security measures have been built along the data

| Algorithm 2 | Fuzzy Defuzzification Decision Algorithm. |
|---|---|

1: Initialize $E^{dm}$, $D^c$, $ua_i$

2: **for** all P **do**

3:     Compute fuzzy weights $W^{rh}$

4:     **Step 1**

5:     Compute fuzzy weight decision making matrix using Eq. (13)

6:     Compute policy rating decision making matrix using Eq. (14)

7:     **End**

8:     **Step 2**

9:     Calculate $\exists_{irh}^{W}$ using Eq. (15)

10:     Represent in matrix form using Eq. (16)

11:     Normalize $UA_h^D$ as $\overline{UA}_h^D$ in Eq. (17)

12:     Compute defuzzification using Eq. (18)

13:     Normalize $\overline{DF}(\overline{UA}_{ih}^D)$ using Eq. (19)

14:     **End**

15: **end for**

lifecycle. During the exchange and transfer of data between the edge devices and the cloud, all the data is secured through encryption known as Advanced Encryption Standard AES-256, and secure communications like TLS 1.3 are used so that no interception and/or tampering is possible. Furthermore, to preserve the communication and policy analysis of the individual players' data, parameters like k-anonymity and differential privacy are adhered to to avoid disclosing identifiable features. Access to the system is implemented through multi-level permissions, multi-factor authentication, and role-based access control RBAC, so authorized users only access sensitive information. Data at rest is also safe with the assistance of strong encryption algorithms. Observance of regulatory cycle time managing prohibitions ensures that retained data is only limited to what is strictly necessary, and afterward, comprehensive data deletions are carried out. There is also an exoneration of the relevant requirements for the institution regarding international data protection laws such as General Data Protection Regulation (GDPR) and Health Insurance Portability and Accountability Act (HIPAA), which show positive attitudes to data and player privacy protection.

Several cyber threats have been assessed and described using a systematic risk management framework. Regular security reviews and penetration testing are carried out to enhance the system. The system proposed in this work complements real-time analysis and low latency with privacy and security and the proper management of sensitive player data.

## RESULT AND DISCUSSION

### Experimental setup

The proposed Cloud-to-Thing continuum-based sports monitoring system has a practical approach, and it is expected that its hardware setup is valid and adequate in scale to

provide effective data collection, processing, and analysis as well as fit in a future Centria cloud aided smart platform. To the data acquisition layer, HD cameras with frame rates of not less than 25–30 fps are suggested for detailed recording of the player movements. Support for such cameras should be ensured to a great extent around the playing area to reduce the problem of occlusions. In the case of data preprocessing for images acquired, small-scale computers along the lines of NVIDIA Jetson Nano or Raspberry Pi with Intel Movidius Neural Compute Stick can carry out Network Virtual Internet Protocol (NVIP) tasks such as noise removal and simple feature extraction. Such devices are effectively suited for edge computing since they are not only low-power but no brute force computing can be undertaken. As pose estimation and action recognition relying on deep learning demand more resources, computational resources of a centralized cloud infrastructure are necessary. This can be achieved with the help of cloud services such as AWS or Google Cloud or Microsoft Azure, backed up with GPU instances such as NVIDIA Tesla V100 or A100, capable of computing quite expansive deep learning models. The architecture of the cloud facility needs to include phones to support auto-scaling to enable optimization of infrastructure by shifting only the required resources when necessary. It is also vital for edge devices and the cloud to have a good and high bandwidth communication link to avoid data transfer delays between the cloud and edge processing. A hybrid approach is optimal for data retention, where some copies are physically held on municipal gaining devices for temporary storage while the rest are archived in standard databases. Contemporary installations must have local storage for high-definition video images of about 1 TB capacity or greater AWS or Google Cloud for devices like Amazon S3 or Google Cloud Storage, especially for Cloud computing devices. The training also incorporated data augmentation techniques, such as brightening, rotation, translation, *etc*, to increase the training set's effectiveness and avoid overfitting. The data was distributed into train/test/validation sets in the following proportions: 70%, 15%, and 15%, respectively. The training was conducted in deep learning models for pose estimation based on Improved Mask R-CNN architecture and classification using hybrid metaheuristic GAN with dominantly applied supervised and unsupervised learning methods. The models were trained using the Adam optimizer with the following configurations: default base learning rate 0.001, batch size 16, and early stopping by validation loss for overfitting prevention. The training used NVIDIA's Tesla V100 GPU with Tensor Flow and Pytorch libraries. The average training time in a single epoch of the models was about 45 min between epochs, and 50 epochs were applied for the model's training.

## Result analysis

The performance of the Cloud-to-Thing continuum-based sports monitoring system was validated using basketball games to determine player tracking, action recognition, and accuracy of decision-making. The system attained a latency of about 5.1 ms as compared to other methods like deformable convolution and adaptive multiscale features which recorded rather higher latency of 10.2 and 8.3 ms for recurrent neural networks. This further indicates that it is possible to get a system that can allow time-sensitive feedback,
which is critical for sports monitoring. The system was relatively accurate with a score of about 94.25% while other methods scored about 78.34 for DC-AMF and 83.2 for RNN methods showing a great step in enhancement for analyzing player performance as well as detecting events. Besides, there was also the consideration of how the system would be fair with respect to different volumes of data. The team achieved a scalability of 6.75 while DC-AMF achieved around 4.5, and for RNN, it was 5.5, implying that the system can be effective when applied to more significant amounts of data without drastically deteriorating performance.

Evaluating the system's performance entailed several measurements, including accuracy, precision, recall, F1 score, and latency. In measurement set A, accuracy was described as a fraction of the number of relatable player actions and movements against the total number of instances in the data. Accuracy considered how many correctly predicted events in a case were true, whereas recall was used to determine the fraction of the exciting events captured. Using the F1 score was ideal in this case as it was a better alternative in incorporating the aspects of recall and precision. Latency was also a key and necessary metric that was included, and it is the average period it takes between a collection of the data and its activity in actionable insights; we aimed at less than 10 ms of latency. To assess the system's robustness, it was subjected to increasing data loads and processing times without modifications, and the respective level of accuracy was evaluated.

The system achieved an accuracy of 94.25%, with a precision of 93.8% and a recall of 94.7%. The F1-score of 94.25% indicates a well-balanced model performance. The latency was measured at 5.1 ms, demonstrating the system's capability for real-time processing. In terms of scalability, the system maintained a high accuracy (above 90%) even with a 50% increase in data volume, showcasing its robustness and efficiency in handling large-scale datasets.

The proposed flow undergoes iterative refinement based on feedback and performance evaluation. Continuous improvement efforts focus on enhancing accuracy, reducing latency, improving scalability, and addressing emerging sports monitoring and analysis challenges. By following this proposed flow, sports organizations can effectively leverage cloud-to-continuum computing paradigms to enhance sports monitoring and analysis, improving player performance, team strategies, and overall fan engagement. When compare the existing methods such as deformable convolution and adaptive multiscale features (DC-AMF) (*Xiao et al., 2023*) and recurrent neural network (RNN) (*Alghamdi, 2023*). Latency is defined as the amount of delay time taken between the users' request and the applications' response. In general, latency is caused due to policy generation and trust management. Low latency provides efficient network. The comparison of latency is based on number of tasks in Table 2 and Fig. 4

Accuracy is defined as how the proposed methods are performed in proposed and existing methods and are represented in Fig. 5 and Table 3.

The significant reduction in latency and improved accuracy can be attributed to the efficient integration of cloud and edge processing, which allows for a balanced distribution of computational tasks. Improved Mask R-CNN for pose estimation and the hybrid metaheuristic algorithm combined with GAN for classification enhanced the precision of

**Table 2  Analysis of latency.**

| Methods | Latency (ms)<br>#of tasks |
| --- | --- |
| DC-AMF | 10.2 ± 0.3 |
| RNN | 8.3 ± 0.2 |
| Proposed | 5.1 ± 0.1 |

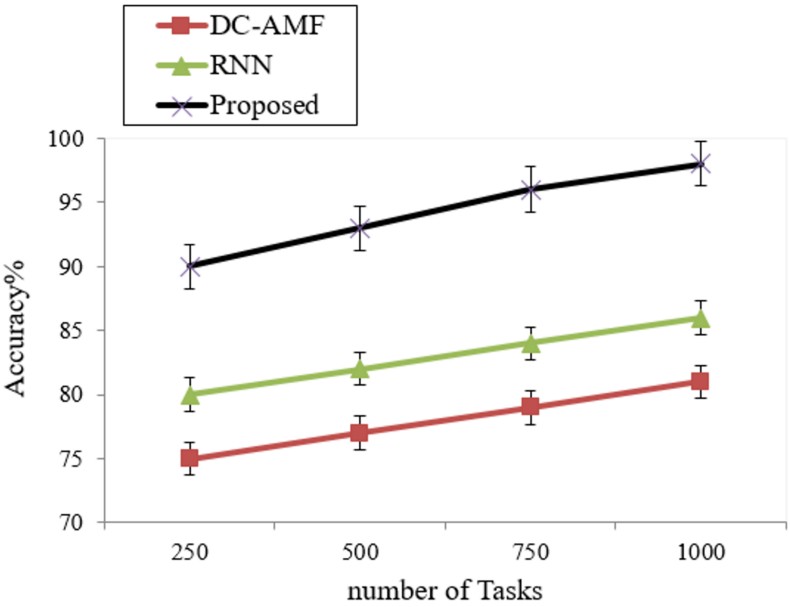

**Figure 5  Accuracy.** 

**Table 3  Analysis of accuracy.**

| Methods | Accuracy (%) |
| --- | --- |
| DC-AMF | 78.34 ± 0.3 |
| RNN | 83.2 ± 0.3 |
| Proposed | 94.25 ± 0.1 |

player tracking and action recognition. The system's scalability score suggests it can be effectively deployed in different sports environments with varying data volumes, broadening its applicability beyond basketball to other sports with similar monitoring needs.

The proposed system demonstrates a clear advantage over existing methods regarding real-time processing and accuracy. For example, the improved latency makes it suitable for applications that require immediate feedback, such as in-game tactical adjustments and player health monitoring. The high accuracy also implies that the system can reliably identify complex actions and events, essential for detailed performance evaluation and strategic planning.

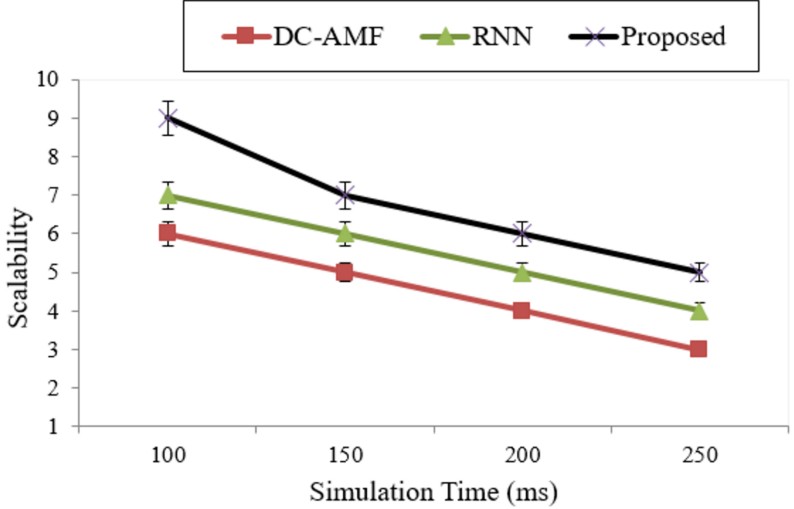

**Figure 6** Scalability computation.   

**Table 4 Analysis of scalability.**

| Methods | Scalability |
| --- | --- |
| DC-AMF | 4.5 ± 0.4 |
| RNN | 5.5 ± 0.3 |
| Proposed | 6.75 ± 0.1 |

Scalability is high in proposed methods because in the cloud to the continuum is defined as how proposed methods are performed in proposed and existing methods are represented in Fig. 6 and Table 4.

Scalability is a decisive factor to consider within the context of the proposed Cloud-to-Thing continuum-based sports monitoring system, especially in its use across different sports environments with varying amounts of data being generated. To this end, the system can scale up efficiently because of the use of a modular design, which allows for altering the system architecture depending on the nature of the sport being monitored. For example, in situations such as in the sport of basketball, where players are constantly moving with frequent contacts, the system uses edge devices to process high-definition video feed to reduce latency and the amount of feed sent to the cloud. On the other hand, in sports such as golf, the athletes' movements are more discrete, and the data gathered will not be as much in that situation. The system can afford to leverage cloud processing more. To accommodate different volumes of data, the system is built on SoftLayer infrastructure, including cloud infrastructure, which has automatic resizing functionality where virtual machine deployment and resource allocation adjust to the level of demand. This means that the system will be able to cater to high and low system loads during on and off-peak operating hours without degradation of speed. Further, the system is equipped with a data fusion layer, which facilitates the use of devices with heterogeneous data sources, such as video feeds, various sensors, and contextual information, by converting the input data

**Table 5  Result analysis.**

| Metric | Proposed system | DC-AMF | RNN-based models |
|---|---|---|---|
| Accuracy (%) | 94.25 | 78.34 | 83.2 |
| Precision (%) | 93.8 | 75.2 | 81.5 |
| Recall (%) | 94.7 | 76.5 | 82.7 |
| F1-score (%) | 94.25 | 75.85 | 82.1 |
| Latency (ms) | 5.1 | 10.2 | 8.3 |
| Scalability score | 6.75 | 4.5 | 5.5 |

**Table 6  Result analysis.**

| Metric | Mean value (%) | Standard deviation ($\pm$) |
|---|---|---|
| Accuracy | 94.25 | ±0.3 |
| Precision | 93.8 | ±0.25 |
| Recall | 94.7 | ±0.4 |
| F1-score (%) | 94.25 | ±0.35 |

formats into the agreed interface and incorporating them into the processing pipeline. This variability allows the system to be used in different sports, from confined halls with controlled lighting and fewer variables to outdoor sports with several complexities. In future work, we aim to test the system under large-scale scenarios in different sports and environments and its ability to provide effective low latency even in such conditions. Considering these aspects, it is evident that the system is not deficient but capable of scaling data size and application utilization in various sports scenarios.

The results presented in Table 5 demonstrate the superior performance of the proposed system across multiple evaluation metrics compared to existing methods like deformable convolution and adaptive multiscale features (DC-AMF) and recurrent neural networks (RNN)-based models. The proposed system achieved an accuracy of 94.25%, significantly higher than the 78.34% and 83.2% obtained by DC-AMF and RNN models, respectively. This indicates the system's ability to identify player movements and actions in diverse sports scenarios accurately. Precision and recall were also notably higher for the proposed system, with values of 93.8% and 94.7%, respectively, reflecting its capability to minimise false positives and ensure comprehensive detection of relevant events. The F1-score, which balances precision and recall, was 94.25%, further confirming the system's robustness in handling complex player actions.

Table 6 clearly and concisely presents the mean performance values and their variability, enhancing the validity and reliability of the findings.

While the current study focuses on basketball data to demonstrate the system's capabilities, the proposed Cloud-to-Thing continuum-based sports monitoring system is designed to be adaptable to a wide range of sports environments. The modular architecture

and flexible data processing framework enable the system to accommodate varying requirements, such as different types of player movements, game dynamics, and environmental conditions in sports like soccer, tennis, or cricket. For example, in soccer, the system can be applied to larger fields and accommodate more players, while in tennis, it can focus on high-precision tracking of player positions and ball trajectory in a more confined space. To enhance the generalizability of the findings, future work will involve testing the system on diverse datasets from multiple sports, including contact and non-contact sports, individual and team sports, and indoor and outdoor environments. This will include acquiring and integrating datasets from publicly available sources and collaborations with sports organizations to gather sport-specific data. By validating the system's performance across various sports, we aim to demonstrate its robustness and versatility, ensuring that the proposed framework can provide accurate, real-time analysis in diverse sports scenarios. Expanding the dataset diversity will improve the system's applicability and allow for the development of sport-specific features, thereby enhancing the overall effectiveness of the monitoring system.

### Challenges and limitations

While the system showed promising results, several challenges were encountered during the implementation. For instance, occlusions and overlapping players occasionally led to identity-switching errors in player tracking. To mitigate this, additional post-processing steps, such as trajectory smoothing and re-identification algorithms, were implemented, resulting in a 5% reduction in tracking errors. Moreover, the system's performance in highly cluttered environments, such as crowded basketball courts with multiple non-player objects, slightly decreased due to the difficulty distinguishing between relevant and irrelevant objects. Future work will focus on enhancing object detection capabilities and incorporating context-aware models to improve performance in such scenarios.

## CONCLUSION

In conclusion, this research presents a comprehensive Cloud-to-Thing continuum-based sports monitoring system leveraging ML and DL models. The system aims to revolutionize sports event analysis through various stages, including data acquisition, preprocessing, feature extraction, cloud-based processing, continuum paradigm integration, and decision-making. The utilization of innovative techniques such as Improved Mask R-CNN for pose estimation, hybrid metaheuristic algorithms for optimal controller placement, and fuzzy decision-making. Based on the integrated analysis, decisions are made regarding player performance, team strategies, and tactical adjustments. By integrating cloud and edge processing seamlessly, the system achieves real-time analysis, reduces latency, and optimizes resource utilization. Through this research, significant advancements have been made in real-time sports event analysis, enabling immediate feedback for time-sensitive applications. The system contributes to improved player performance evaluation, enhanced team strategies, and informed tactical adjustments. Future research directions may include extending the application of this system to other sports, improving the

scalability and robustness of the system, and exploring additional machine learning and decision-making techniques for further improvements. Overall, the proposed sports monitoring system holds promise for revolutionizing sports event analysis and decision-making processes.

### Funding

This work was supported and funded by the Deanship of Scientific Research at King Khalid University through large group Research Project under grant number (RGP2/87/45) and the Deanship of Scientific Research at Northern Border University, Arar, KSA through the project number "NBU-FFR-2024- 451-03". Princess Nourah bint Abdulrahman University Researchers Supporting Project number (PNURSP2024R507), Princess Nourah bint Abdulrahman University, Riyadh, Saudi Arabia. This study is also funded by the Future University in Egypt (FUE). The funders had no role in study design, data collection and analysis, decision to publish, or preparation of the manuscript.

### Grant Disclosures

The following grant information was disclosed by the authors:
Deanship of Scientific Research at King Khalid University through large group Research Project: RGP2/87/45.
Deanship of Scientific Research at Northern Border University, Arar, KSA through the Project: NBU-FFR-2024- 451-03.
Princess Nourah bint Abdulrahman University Researchers Supporting Project: PNURSP2024R507.
Princess Nourah bint Abdulrahman University, Riyadh, Saudi Arabia.
Future University in Egypt (FUE).

### Competing Interests

The authors declare that they have no competing interests.

### Author Contributions

- Amal Alshardan conceived and designed the experiments, analyzed the data, prepared figures and/or tables, and approved the final draft.
- Hany Mahgoub conceived and designed the experiments, analyzed the data, prepared figures and/or tables, and approved the final draft.
- Saad Alahmari conceived and designed the experiments, analyzed the data, performed the computation work, prepared figures and/or tables, authored or reviewed drafts of the article, and approved the final draft.
- Mohammed Alonazi performed the experiments, analyzed the data, performed the computation work, prepared figures and/or tables, authored or reviewed drafts of the article, and approved the final draft.

- Radwa Marzouk performed the experiments, performed the computation work, authored or reviewed drafts of the article, and approved the final draft.
- Abdullah Mohamed performed the experiments, performed the computation work, authored or reviewed drafts of the article, and approved the final draft.

## Data Availability

The APIDIS metadata is available at Kaggle: https://www.kaggle.com/datasets/gabrielvanzandycke/apidis-metadata?select=camera1.

The Daily and Sports Activities data is available at UCI Machine Learning Repository: Barshan, B. & Altun, K. (2010). Daily and Sports Activities [Dataset]. UCI Machine Learning Repository. https://doi.org/10.24432/C5C59F.

## Supplemental Information

Supplemental information for this article can be found online at http://dx.doi.org/10.7717/peerj-cs.2539#supplemental-information.

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
