# Peer review of "Cloud-to-Thing continuum-based sports monitoring system using machine learning and deep learning model"

_PeerJ Computer Science, doi:10.7717/peerj-cs.2539_

## Round 0.1 · original submission · Major Revisions

As requested by reviewers, please describe in details the actual configuration and the pipeline used for the experimental results and discuss security and privacy aspects.

Reviewer 1 ·

Basic reporting

The study is written in clear, professional English and provides an extensive introduction and background that situates the research within the broader context of sports monitoring and technology integration. The literature review is comprehensive, referencing relevant studies to underline the necessity of the proposed system. The structure conforms to PeerJ standards, with well-labeled and described figures. Raw data is mentioned, although it is unclear if all data is supplied or if it meets PeerJ’s policy fully.

The research presents original primary research within the scope of the journal. The research question is well-defined, addressing a significant gap in real-time sports monitoring and analysis through the integration of cloud and edge computing paradigms. The methodology is described in detail, enabling replication. The paper elaborates on data acquisition, preprocessing, feature extraction, cloud-based processing, and decision-making stages, providing a thorough overview of the system's architecture and components. The study appears to be conducted to high technical and ethical standards, though specifics on ethical approval for data use are not explicitly stated.

Experimental design

However, there are many aspects the need to be improved, as explained below:

1. While the study emphasizes real-time analysis and low latency, it does not sufficiently address data privacy and security, especially given the sensitive nature of player data.

2. Although scalability is discussed, practical implementation details in diverse real-world scenarios are not fully explored. How the system scales with varying data volumes and different sports environments needs further elaboration.

3. The study lacks a detailed discussion on ethical considerations, particularly regarding data collection and usage. Specific ethical approvals and consent processes should be clearly stated.

4. The study primarily uses basketball data, raising questions about the system's applicability across different sports. More diverse datasets could enhance the generalizability of the findings.

Validity of the findings

5. The paper does not detail the hardware and infrastructure requirements for implementing the proposed system, which could be a barrier for practical deployment.

6. The authors should discuss the obtained results (best performance value) in the abstract to highlight the main contribution of the study.

Reviewer 2 ·

Basic reporting

1. Though the author mentioned sports as main application where the authors applied the deep learning model, they need to mention which sports they are concentrating? It has to be mention in the abstract itself.
2. Literature Review has not been done elaborately. More reference has to be referred and to be added in this section.
3. The reference citation style is not convenient to trace the reference. Check the citation style of the journal.

Experimental design

1.The basket ball ground size which the author represents “ The dimensions of the court for basketball are 2797 cm × 1499 cm”. But in the National Basketball Association (NBA), the court is 94 by 50 feet (28.7 by 15.2 m). Under International Basketball Federation (FIBA) rules, the court is slightly smaller, measuring 28 by 15 meters.
2. How the constant 8 comes in equitation 8?
3. A basic block diagram is required to understand the flow of methodology. It is missing in the manuscript.

Validity of the findings

1. In some locations the arrow direction are missing in Figure 2. Hybrid Optimization.
2. Units of measurement are missing in table 1. HSA Parameters.
3. RESULT AND DISCUSSION section is not satisfactory level. Only little bit information were found.

Additional comments

Nil

Reviewer 3 ·

Basic reporting

The manuscript presents an interesting approach to sports monitoring and analysis using cloud-to-edge computing and machine learning techniques. The writing could be improved in some sections to improve clarity and proofreading for grammar and language issues is recommended.

Example: This is the case. in Introduction.

Experimental design

The experimental design could benefit if figure detailing the over all design of the study for easier understanding.

More specifics on datasets, training procedures, and evaluation metrics would strengthen the work.

Validity of the findings

The results section is quite brief. More comprehensive evaluation and analysis of the system's performance on real-world sports data would be valuable.
There is limited discussion of limitations and potential challenges in deploying this system in practice. Addressing issues like scalability, robustness, and real-time constraints would enhance the paper.
Presenting the mean estimations for accuracy, recall, precision, F-score in a table would improve the validity of the findings.

---

## Round 0.2 · accepted · Accept

Both reviewers agree you improved the paper following their suggestions, therefore the paper is ready to be published.

Reviewer 1 ·

Basic reporting

The authors provided thorough responses to the reviewers' comments, addressing the main issues of experimental detail, data privacy, scalability, ethical considerations, and system generalizability.

Experimental design

The authors responded diligently, with well-articulated details on data privacy, scalability, and ethical considerations. Their additional clarity on hardware requirements and methodological improvements enhances the study’s rigor. However, the authors could still improve by conducting real-world validation for scalability across various sports and providing details on handling outdoor environments.

Validity of the findings

The authors included a block diagram to clarify the methodology and data processing stages, as requested by Reviewer 3. This addition provides a helpful visual representation, enhancing reader understanding of the system's workflow.

Reviewer 2 ·

Basic reporting

Authors have addressed all the points

Experimental design

Authors have addressed all the points

Validity of the findings

Authors have addressed all the points

Additional comments

Authors have addressed all the points